# Are You Talking to a Machine?
# Dataset and Methods for Multilingual Image Question Answering

**Haoyuan Gao**[1] **Junhua Mao**[2] **Jie Zhou**[1] **Zhiheng Huang**[1] **Lei Wang**[1] **Wei Xu**[1]

[1]Baidu Research     [2]University of California, Los Angeles

gaohaoyuan@baidu.com,mjhustc@ucla.edu,{zhoujie01,huangzhiheng,wanglei22,wei.xu}@baidu.com

## Abstract

In this paper, we present the *mQA* model, which is able to answer questions about the content of an image. The answer can be a sentence, a phrase or a single word. Our model contains four components: a Long Short-Term Memory (LSTM) to extract the question representation, a Convolutional Neural Network (CNN) to extract the visual representation, an LSTM for storing the linguistic context in an answer, and a fusing component to combine the information from the first three components and generate the answer. We construct a Freestyle Multilingual Image Question Answering (*FM-IQA*) dataset to train and evaluate our mQA model. It contains over 150,000 images and 310,000 freestyle Chinese question-answer pairs and their English translations. The quality of the generated answers of our mQA model on this dataset is evaluated by human judges through a *Turing Test*. Specifically, we mix the answers provided by humans and our model. The human judges need to distinguish our model from the human. They will also provide a score (i.e. 0, 1, 2, the larger the better) indicating the quality of the answer. We propose strategies to monitor the quality of this evaluation process. The experiments show that in 64.7% of cases, the human judges cannot distinguish our model from humans. The average score is 1.454 (1.918 for human). The details of this work, including the FM-IQA dataset, can be found on the project page: http://idl.baidu.com/FM-IQA.html.

## 1 Introduction

Recently, there is increasing interest in the field of multimodal learning for both natural language and vision. In particular, many studies have made rapid progress on the task of image captioning [26, 15, 14, 40, 6, 8, 4, 19, 16, 42]. Most of them are built based on deep neural networks (e.g. deep Convolutional Neural Networks (CNN [17]), Recurrent Neural Network (RNN [7]) or Long Short-Term Memory (LSTM [12])). The large-scale image datasets with sentence annotations (e.g., [21, 43, 11]) play a crucial role in this progress. Despite the success of these methods, there are still many issues to be discussed and explored. In particular, the task of image captioning only requires generic sentence descriptions of an image. But in many cases, we only care about a particular part or object of an image. The image captioning task lacks the interaction between the computer and the user (as we cannot input our preference and interest).

In this paper, we focus on the task of visual question answering. In this task, the method needs to provide an answer to a freestyle question about the content of an image. We propose the mQA model to address this task. The inputs of the model are an image and a question. This model has four components (see Figure 2). The first component is an LSTM network that encodes a natural language sentence into a dense vector representation. The second component is a deep Convolutional Neural Network [36] that extracted the image representation. This component was pre-trained on ImageNet Classification Task [33] and is fixed during the training. The third component is another LSTM network that encodes the information of the current word and previous words in the answer into dense representations. The fourth component fuses the information from the first three components to predict the next word in the answer. We jointly train the first, third and fourth components by maximizing the probability of the groundtruth answers in the training set using a log-likelihood loss

| | | | | | |
|---|---|---|---|---|---|
| Image | 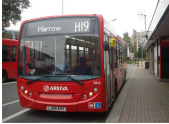 |  |  |  |  |
| Question | 公共汽车是什么颜色的? <br> What is the color of the bus? | 黄色的是什么? <br> What is there in yellow? | 草地上除了人以外还有什么动物? <br> What is there on the grass, except the person? | 猫咪在哪里? <br> Where is the kitty? | 观察一下说出食物里任意一种蔬菜的名字? <br> Please look carefully and tell me what is the name of the vegetables in the plate? |
| Answer | 公共汽车是红色的。 <br> The bus is red. | 香蕉。 <br> Bananas. | 羊。 <br> Sheep. | 在椅子上 。 <br> On the chair. | 西兰花 。 <br> Broccoli. |

Figure 1: Sample answers to the visual question generated by our model on the newly proposed Freestyle Multilingual Image Question Answering (FM-IQA) dataset.

function. To lower down the risk of overfitting, we allow the weight sharing of the word embedding layer between the LSTMs in the first and third components. We also adopt the transposed weight sharing scheme as proposed in [25], which allows the weight sharing between word embedding layer and the fully connected Softmax layer.

To train our method, we construct a large-scale Freestyle Multilingual Image Question Answering dataset[1] (FM-IQA, see details in Section 4) based on the MS COCO dataset [21]. The current version of the dataset contains 158,392 images with 316,193 Chinese question-answer pairs and their corresponding English translations.[2] To diversify the annotations, the annotators are allowed to raise any question related to the content of the image. We propose strategies to monitor the quality of the annotations. This dataset contains a wide range of AI related questions, such as action recognition (e.g., "Is the man trying to buy vegetables?"), object recognition (e.g., "What is there in yellow?"), positions and interactions among objects in the image (e.g. "Where is the kitty?") and reasoning based on commonsense and visual content (e.g. "Why does the bus park here?", see last column of Figure 3).

Because of the variability of the freestyle question-answer pairs, it is hard to accurately evaluate the method with automatic metrics. We conduct a Visual Turing Test [38] using human judges. Specifically, we mix the question-answer pairs generated by our model with the same set of question-answer pairs labeled by annotators. The human judges need to determine whether the answer is given by a model or a human. In addition, we also ask them to give a score of 0 (i.e. wrong), 1 (i.e. partially correct), or 2 (i.e. correct). The results show that our mQA model passes 64.7% of this test (treated as answers of a human) and the average score is 1.454. In the discussion, we analyze the failure cases of our model and show that combined with the m-RNN [24] model, our model can automatically ask a question about an image and answer that question.

## 2 Related Work

Recent work has made significant progress using deep neural network models in both the fields of computer vision and natural language. For computer vision, methods based on Convolutional Neural Network (CNN [20]) achieve the state-of-the-art performance in various tasks, such as object classification [17, 34, 17], detection [10, 44] and segmentation [3]. For natural language, the Recurrent Neural Network (RNN [7, 27]) and the Long Short-Term Memory network (LSTM [12]) are also widely used in machine translation [13, 5, 35] and speech recognition [28].

The structure of our mQA model is inspired by the m-RNN model [24] for the image captioning and image-sentence retrieval tasks. It adopts a deep CNN for vision and a RNN for language. We extend the model to handle the input of question and image pairs, and generate answers. In the experiments, we find that we can learn how to ask a good question about an image using the m-RNN model and this question can be answered by our mQA model.

There has been recent effort on the visual question answering task [9, 2, 22, 37]. However, most of them use a pre-defined and restricted set of questions. Some of these questions are generated from a template. In addition, our FM-IQA dataset is much larger than theirs (e.g., there are only 2591 and 1449 images for [9] and [22] respectively).

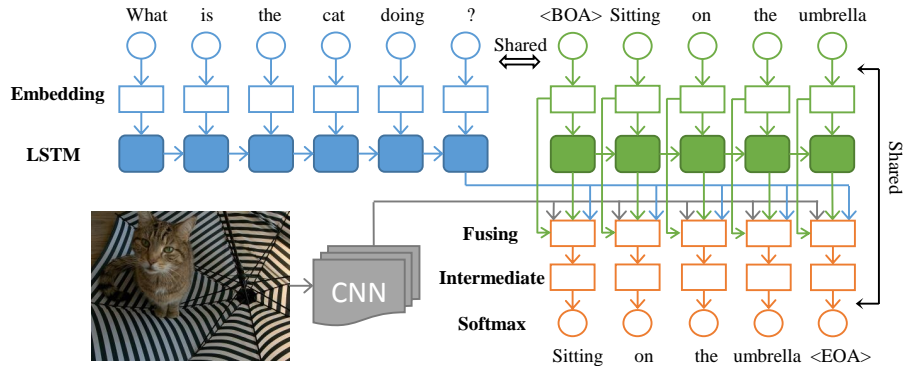

What is the cat doing ? &lt;BOA&gt; Sitting on the umbrella

Figure 2: Illustration of the mQA model architecture. We input an image and a question about the image (i.e. "What is the cat doing?") to the model. The model is trained to generate the answer to the question (i.e. "Sitting on the umbrella"). The weight matrix in the word embedding layers of the two LSTMs (one for the question and one for the answer) are shared. In addition, as in [25], this weight matrix is also shared, in a transposed manner, with the weight matrix in the Softmax layer. Different colors in the figure represent different components of the model. (Best viewed in color.)

There are some concurrent and independent works on this topic: [1, 23, 32]. [1] propose a large-scale dataset also based on MS COCO. They also provide some simple baseline methods on this dataset. Compared to them, we propose a stronger model for this task and evaluate our method using human judges. Our dataset also contains two different kinds of language, which can be useful for other tasks, such as machine translation. Because we use a different set of annotators and different requirements of the annotation, our dataset and the [1] can be complementary to each other, and lead to some interesting topics, such as dataset transferring for visual question answering.

Both [23] and [32] use a model containing a single LSTM and a CNN. They concatenate the question and the answer (for [32], the answer is a single word. [23] also prefer a single word as the answer), and then feed them to the LSTM. Different from them, we use two separate LSTMs for questions and answers respectively in consideration of the different properties (e.g. grammar) of questions and answers, while allow the sharing of the word-embeddings. For the dataset, [23] adopt the dataset proposed in [22], which is much smaller than our FM-IQA dataset. [32] utilize the annotations in MS COCO and synthesize a dataset with four pre-defined types of questions (i.e. object, number, color, and location). They also synthesize the answer with a single word. Their dataset can also be complementary to ours.

## 3 The Multimodal QA (mQA) Model

We show the architecture of our mQA model in Figure 2. The model has four components: (I). a Long Short-Term Memory (LSTM [12]) for extracting semantic representation of a question, (II). a deep Convolutional Neural Network (CNN) for extracting the image representation, (III). an LSTM to extract representation of the current word in the answer and its linguistic context, and (IV). a fusing component that incorporates the information from the first three parts together and generates the next word in the answer. These four components can be jointly trained together [3]. The details of the four model components are described in Section 3.1. The effectiveness of the important components and strategies are analyzed in Section 5.3.

The inputs of the model are a question and the reference image. The model is trained to generate the answer. The words in the question and answer are represented by one-hot vectors (i.e. binary vectors with the length of the dictionary size $N$ and have only one non-zero vector indicating its index in the word dictionary). We add a $\langle BOA \rangle$ sign and a $\langle EOA \rangle$ sign, as two spatial words in the word dictionary, at the beginning and the end of the training answers respectively. They will be used for generating the answer to the question in the testing stage.

In the testing stage, we input an image and a question about the image into the model first. To generate the answer, we start with the start sign $\langle BOA \rangle$ and use the model to calculate the probability distribution of the next word. We then use a beam search scheme that keeps the best $K$ candidates

with the maximum probabilities according to the Softmax layer. We repeat the process until the model generates the end sign of the answer $\langle BOA \rangle$.

## 3.1 The Four Components of the mQA Model

(I). The semantic meaning of the question is extracted by the first component of the model. It contains a 512 dimensional word embedding layer and an LSTM layer with 400 memory cells. The function of the word embedding layer is to map the one-hot vector of the word into a dense semantic space. We feed this dense word representation into the LSTM layer.

LSTM [12] is a Recurrent Neural Network [7] that is designed for solving the gradient explosion or vanishing problem. The LSTM layer stores the context information in its memory cells and serves as the bridge among the words in a sequence (e.g. a question). To model the long term dependency in the data more effectively, LSTM add three gate nodes to the traditional RNN structure: the input gate, the output gate and the forget gate. The input gate and output gate regulate the read and write access to the LSTM memory cells. The forget gate resets the memory cells when their contents are out of date. Different from [23, 32], the image representation does not feed into the LSTM in this component. We believe this is reasonable because questions are just another input source for the model, so we should not add images as the supervision for them. The information stored in the LSTM memory cells of the last word in the question (i.e. the question mark) will be treated as the representation of the sentence.

(II). The second component is a deep Convolutional Neural Network (CNN) that generates the representation of an image. In this paper, we use the GoogleNet [36]. Note that other CNN models, such as AlexNet [17] and VggNet [34], can also be used as the component in our model. We remove the final SoftMax layer of the deep CNN and connect the remaining top layer to our model.

(III). The third component also contains a word embedding layer and an LSTM. The structure is similar to the first component. The activation of the memory cells for the words in the answer, as well as the word embeddings, will be fed into the fusing component to generate the next words in the answer.

In [23, 32], they concatenate the training question and answer, and use a single LSTM. Because of the different properties (i.e. grammar) of question and answer, in this paper, we use two separate LSTMs for questions and answers respectively. We denote the LSTMs for the question and the answer as LSTM(Q) and LSTM(A) respectively in the rest of the paper. The weight matrix in LSTM(Q) is not shared with the LSTM(A) in the first components. Note that the semantic meaning of single words should be the same for questions and answers so that we share the parameters in the word-embedding layer for the first and third component.

(IV). Finally, the fourth component fuses the information from the first three layers. Specifically, the activation of the fusing layer $\mathbf{f}(t)$ for the $t^{th}$ word in the answer can be calculated as follows:

$$\mathbf{f}(t) = g(\mathbf{V}_{r_Q}\mathbf{r}_Q + \mathbf{V}_I\mathbf{I} + \mathbf{V}_{r_A}\mathbf{r}_A(t) + \mathbf{V}_w\mathbf{w}(t));\tag{1}$$

where "+" denotes element-wise addition, $\mathbf{r}_Q$ stands for the activation of the LSTM(Q) memory cells of the last word in the question, $\mathbf{I}$ denotes the image representation, $\mathbf{r}_A(t)$ and $\mathbf{w}(t)$ denotes the activation of the LSTM(A) memory cells and the word embedding of the $t^{th}$ word in the answer respectively. $\mathbf{V}_{r_Q}$, $\mathbf{V}_I$, $\mathbf{V}_{r_A}$, and $\mathbf{V}_w$ are the weight matrices that need to be learned. $g(.)$ is an element-wise non-linear function.

After the fusing layer, we build an intermediate layer that maps the dense multimodal representation in the fusing layer back to the dense word representation. We then build a fully connected Softmax layer to predict the probability distribution of the next word in the answer. This strategy allows the weight sharing between word embedding layer and the fully connected Softmax layer as introduced in [25] (see details in Section 3.2).

Similar to [25], we use the sigmoid function as the activation function of the three gates and adopt ReLU [30] as the non-linear function for the LSTM memory cells. The non-linear activation function for the word embedding layer, the fusing layer and the intermediate layer is the scaled hyperbolic tangent function [20]: $g(x) = 1.7159 \cdot \tanh(\frac{2}{3}x)$.

## 3.2 The Weight Sharing Strategy

As mentioned in Section 2, our model adopts different LSTMs for the question and the answer because of the different grammar properties of questions and answers. However, the meaning of

single words in both questions and answers should be the same. Therefore, we share the weight matrix between the word-embedding layers of the first component and the third component.

In addition, this weight matrix for the word-embedding layers is shared with the weight matrix in the fully connected Softmax layer in a transposed manner. Intuitively, the function of the weight matrix in the word-embedding layer is to encode the one-hot word representation into a dense word representation. The function of the weight matrix in the Softmax layer is to decode the dense word representation into a pseudo one-word representation, which is the inverse operation of the word-embedding. This strategy will reduce nearly half of the parameters in the model and is shown to have better performance in image captioning and novel visual concept learning tasks [25].

### 3.3 Training Details

The CNN we used is pre-trained on the ImageNet classification task [33]. This component is fixed during the QA training. We adopt a log-likelihood loss defined on the word sequence of the answer. Minimizing this loss function is equivalent to maximizing the probability of the model to generate the groundtruth answers in the training set. We jointly train the first, second and the fourth components using stochastic gradient decent method. The initial learning rate is 1 and we decrease it by a factor of 10 for every epoch of the data. We stop the training when the loss on the validation set does not decrease within three epochs. The hyperparameters of the model are selected by cross-validation.

For the Chinese question answering task, we segment the sentences into several word phrases. These phrases can be treated equivalently to the English words.

## 4 The Freestyle Multilingual Image Question Answering (FM-IQA) Dataset

Our method is trained and evaluated on a large-scale multilingual visual question answering dataset. In Section 4.1, we will describe the process to collect the data, and the method to monitor the quality of annotations. Some statistics and examples of the dataset will be given in Section 4.2. The latest dataset is available on the project page: `http://idl.baidu.com/FM-IQA.html`

### 4.1 The Data Collection

We start with the 158,392 images from the newly released MS COCO [21] training, validation and testing set as the initial image set. The annotations are collected using Baidu's online crowdsourcing server[4]. To make the labeled question-answer pairs diversified, the annotators are free to give any type of questions, as long as these questions are related to the content of the image. The question should be answered by the visual content and commonsense (e.g., we are not expecting to get questions such as "What is the name of the person in the image?"). The annotators need to give an answer to the question themselves.

On the one hand, the freedom we give to the annotators is beneficial in order to get a freestyle, interesting and diversified set of questions. On the other hand, it makes it harder to control the quality of the annotation compared to a more detailed instruction. To monitor the annotation quality, we conduct an initial quality filtering stage. Specifically, we randomly sampled 1,000 images as a quality monitoring dataset from the MS COCO dataset as an initial set for the annotators (they do not know this is a test). We then sample some annotations and rate their quality after each annotator finishes some labeling on this quality monitoring dataset (about 20 question-answer pairs per annotator). We only select a small number of annotators (195 individuals) whose annotations are satisfactory (i.e. the questions are related to the content of the image and the answers are correct). We also give preference to the annotators who provide interesting questions that require high level reasoning to give the answer. Only the selected annotators are permitted to label the rest of the images. We pick a set of good and bad examples of the annotated question-answer pairs from the quality monitoring dataset, and show them to the selected annotators as references. We also provide reasons for selecting these examples. After the annotation of all the images is finished, we further refine the dataset and remove a small portion of the images with badly labeled questions and answers.

### 4.2 The Statistics of the Dataset

Currently there are 158,392 images with 316,193 Chinese question-answer pairs and their English translations. Each image has at least two question-answer pairs as annotations. The average lengths

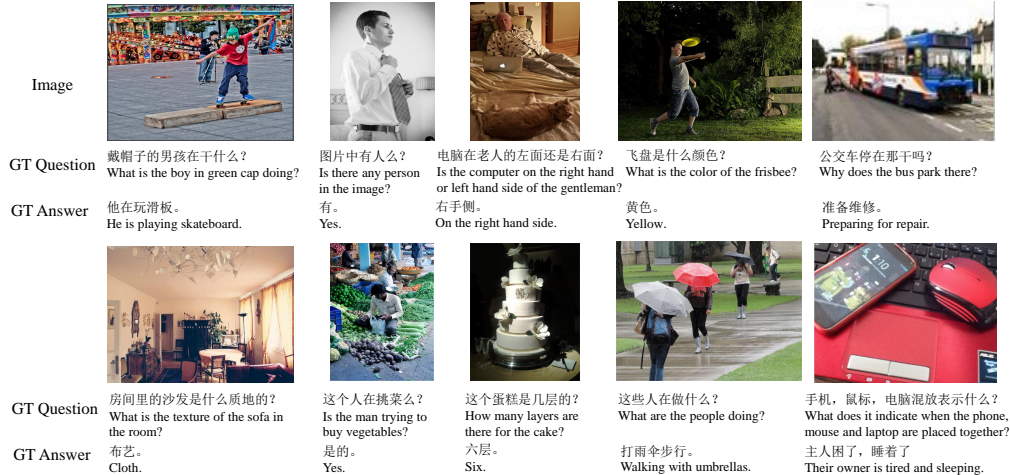

| | | | | |
|---|---|---|---|---|
| **Image** | | | | |
| **GT Question** | 戴帽子的男孩在干什么？ <br> What is the boy in green cap doing? | 图片中有人么？ <br> Is there any person in the image? | 电脑在老人的左面还是右面？ <br> Is the computer on the right hand or left hand side of the gentleman? | 飞盘是什么颜色？ <br> What is the color of the frisbee? | 公交车停在那干吗？ <br> Why does the bus park there? |
| **GT Answer** | 他在玩滑板。 <br> He is playing skateboard. | 有。 <br> Yes. | 右手侧。 <br> On the right hand side. | 黄色。 <br> Yellow. | 准备维修。 <br> Preparing for repair. |

| | | | | |
|---|---|---|---|---|
| **GT Question** | 房间里的沙发是什么质地的？ <br> What is the texture of the sofa in the room? | 这个人在挑菜么？ <br> Is the man trying to buy vegetables? | 这个蛋糕是几层的？ <br> How many layers are there for the cake? | 这些人在做什么？ <br> What are the people doing? | 手机，鼠标，电脑混放表示什么？ <br> What does it indicate when the phone, mouse and laptop are placed together? |
| **GT Answer** | 布艺。 <br> Cloth. | 是的。 <br> Yes. | 六层。 <br> Six. | 打雨伞步行。 <br> Walking with umbrellas. | 主人困了，睡着了 <br> Their owner is tired and sleeping. |

Figure 3: Sample images in the FM-IQA dataset. This dataset contains 316,193 Chinese question-answer pairs with corresponding English translations.

of the questions and answers are 7.38 and 3.82 respectively measured by Chinese words. Some sample images are shown in Figure 3. We randomly sampled 1,000 question-answer pairs and their corresponding images as the test set.

The questions in this dataset are diversified, which requires a vast set of AI capabilities in order to answer them. They contain some relatively simple image understanding questions of, e.g., the actions of objects (e.g., "What is the boy in green cap doing?"), the object class (e.g., "Is there any person in the image?"), the relative positions and interactions among objects (e.g., "Is the computer on the right or left side of the gentleman?"), and the attributes of the objects (e.g., "What is the color of the frisbee?"). In addition, the dataset contains some questions that need a high-level reasoning with clues from vision, language and commonsense. For example, to answer the question of "Why does the bus park there?", we should know that this question is about the parked bus in the image with two men holding tools at the back. Based on our commonsense, we can guess that there might be some problems with the bus and the two men in the image are trying to repair it. These questions are hard to answer but we believe they are actually the most interesting part of the questions in the dataset. We categorize the questions into 8 types and show the statistics of them on the project page.

The answers are also diversified. The annotators are allowed to give a single phrase or a single word as the answer (e.g. "Yellow") or, they can give a complete sentence (e.g. "The frisbee is yellow").

## 5 Experiments

For the very recent works for visual question answering ([32, 23]), they test their method on the datasets where the answer of the question is a single word or a short phrase. Under this setting, it is plausible to use automatic evaluation metrics that measure the single word similarity, such as Wu-Palmer similarity measure (WUPS) [41]. However, for our newly proposed dataset, the answers in the dataset are freestyle and can be complete sentences. For most of the cases, there are numerous choices of answers that are all correct. The possible alternatives are BLEU score [31], METEOR [18], CIDEr [39] or other metrics that are widely used in the image captioning task [24]. The problem of these metrics is that there are only a few words in an answer that are semantically critical. These metrics tend to give equal weights (e.g. BLEU and METEOR) or different weights according to the tf-idf frequency term (e.g. CIDEr) of the words in a sentence, hence cannot fully show the importance of the keywords. The evaluation of the image captioning task suffers from the same problem (not as severe as question answering because it only needs a general description).

To avoid these problems, we conduct a real *Visual Turing Test* using human judges for our model, which will be described in details in Section 5.1. In addition, we rate each generated sentences with a score (the larger the better) in Section 5.2, which gives a more fine-grained evaluation of our method. In Section 5.3, we provide the performance comparisons of different variants of our mQA model on the validation set.

|  | Visual Turing Test | | | Human Rated Scores | | | |
|---|---|---|---|---|---|---|---|
|  | Pass | Fail | Pass Rate (%) | 2 | 1 | 0 | Avg. Score |
| Human | 948 | 52 | 94.8 | 927 | 64 | 9 | 1.918 |
| blind-QA | 340 | 660 | 34.0 | - | - | - | - |
| mQA | 647 | 353 | 64.7 | 628 | 198 | 174 | 1.454 |

Table 1: The results of our mQA model for our FM-IQA dataset.

## 5.1 The Visual Turing Test

In this Visual Turing Test, a human judge will be presented with an image, a question and the answer to the question generated by the testing model or by human annotators. He or she need to determine, based on the answer, whether the answer is given by a human (i.e. pass the test) or a machine (i.e. fail the test).

In practice, we use the images and questions from the test set of our FM-IQA dataset. We use our mQA model to generate the answer for each question. We also implement a baseline model of the question answering without visual information. The structure of this baseline model is similar to mQA, except that we do not feed the image information extracted by the CNN into the fusing layer. We denote it as blind-QA. The answers generated by our mQA model, the blind-QA model and the groundtruth answer are mixed together. This leads to 3000 question answering pairs with the corresponding images, which will be randomly assigned to 12 human judges.

The results are shown in Table 1. It shows that 64.7% of the answers generated by our mQA model are treated as answers provided by a human. The blind-QA performs very badly in this task. But some of the generated answers pass the test. Because some of the questions are actually multi-choice questions, it is possible to get a correct answer by random guess based on pure linguistic clues.

To study the variance of the VTT evaluation across different sets of human judges, we conduct two additional evaluations with different groups of judges under the same setting. The standard deviations of the passing rate are 0.013, 0.019 and 0.024 for human, the blind-mQA model and mQA model respectively. It shows that VTT is a stable and reliable evaluation metric for this task.

## 5.2 The Score of the Generated Answer

The Visual Turing Test only gives a rough evaluation of the generated answers. We also conduct a fine-grained evaluation with scores of "0", "1", or "2". "0" and "2" mean that the answer is totally wrong and perfectly correct respectively. "1" means that the answer is only partially correct (e.g., the general categories are right but the sub-categories are wrong) and makes sense to the human judges. The human judges for this task are not necessarily the same people for the Visual Turing Test. After collecting the results, we find that some human judges also rate an answer with "1" if the question is very hard to answer so that even a human, without carefully looking at the image, will possibly make mistakes. We show randomly sampled images whose scores are "1" in Figure 4.

The results are shown in Table 1. We show that among the answers that are not perfectly correct (i.e. scores are not 2), over half of them are partially correct. Similar to the VTT evaluation process, we also conducts two additional groups of this scoring evaluation. The standard deviations of human and our mQA model are 0.020 and 0.041 respectively. In addition, for 88.3% and 83.9% of the cases, the three groups give the same score for human and our mQA model respectively.

## 5.3 Performance Comparisons of the Different mQA Variants

In order to show the effectiveness of the different components and strategies of our mQA model, we implement three variants of the mQA in Figure 2. For the first variant (i.e. "mQA-avg-question"), we replace the first LSTM component of the model (i.e. the LSTM to extract the question embedding)

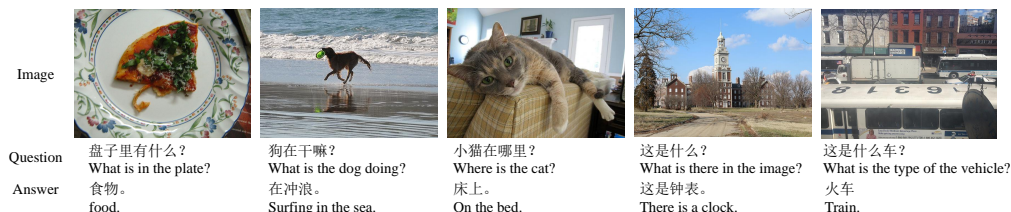

Figure 4: Random examples of the answers generated by the mQA model with score "1" given by the human judges.

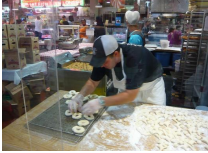   

| Image | | | | |
|---|---|---|---|---|
| Generated Question | 这是在什么地方？<br>Where is this? | 这个人在打网球么？<br>Is this guy playing tennis? | 这是什么食物？<br>What kind of food is this? | 电脑在哪里？<br>Where is the computer? |
| Answer | 这是在厨房。<br>This is the kitchen room. | 是的。<br>Yes. | 披萨。<br>Pizza. | 在桌子上。<br>On the desk. |

Figure 5: The sample generated questions by our model and their answers.

with the average embedding of the words in the question using word2vec [29]. It is used to show the effectiveness of the LSTM as a question embedding learner and extractor. For the second variant (i.e. "mQA-same-LSTMs"), we use two shared-weights LSTMs to model question and answer. It is used to show the effectiveness of the decoupling strategy of the weights of the LSTM(Q) and the LSTM(A) in our model. For the third variant (i.e. "mQA-noTWS"), we do not adopt the Transposed Weight Sharing (TWS) strategy. It is used to show the effectiveness of TWS.

| | Word Error | Loss |
|---|---|---|
| mQA-avg-question | 0.442 | 2.17 |
| mQA-same-LSTMs | 0.439 | 2.09 |
| mQA-noTWS | 0.438 | 2.14 |
| mQA-complete | **0.393** | **1.91** |

Table 2: Performance comparisons of the different mQA variants.

The word error rates and losses of the three variants and the complete mQA model (i.e. mQA-complete) are shown in Table 2. All of the three variants performs worse than our mQA model.

## 6  Discussion

In this paper, we present the mQA model, which is able to give a sentence or a phrase as the answer to a freestyle question for an image. To validate the effectiveness of the method, we construct a Freestyle Multilingual Image Question Answering (FM-IQA) dataset containing over 310,000 question-answer pairs. We evaluate our method using human judges through a real Turing Test. It shows that 64.7% of the answers given by our mQA model are treated as the answers provided by a human. The FM-IQA dataset can be used for other tasks, such as visual machine translation, where the visual information can serve as context information that helps to remove ambiguity of the words in a sentence.

We also modified the LSTM in the first component to the multimodal LSTM shown in [25]. This modification allows us to generate a free-style question about the content of image, and provide an answer to this question. We show some sample results in Figure 5.

We show some failure cases of our model in Figure 6. The model sometimes makes mistakes when the commonsense reasoning through background scenes is incorrect (e.g., for the image in the first column, our method says that the man is surfing but the small yellow frisbee in the image indicates that he is actually trying to catch the frisbee. It also makes mistakes when the targeting object that the question focuses on is too small or looks very similar to other objects (e.g. images in the second and fourth column). Another interesting example is the image and question in the fifth column of Figure 6. Answering this question is very hard since it needs high level reasoning based on the experience from everyday life. Our model outputs a $\langle OOV \rangle$ sign, which is a special word we use when the model meets a word which it has not seen before (i.e. does not appear in its word dictionary).

In future work, we will try to address these issues by incorporating more visual and linguistic information (e.g. using object detection or using attention models).

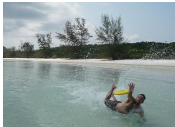    

| Image | | | | | |
|---|---|---|---|---|---|
| Question | 帅哥在干什么？<br>What is the handsome boy doing? | 这是什么？<br>What is there in the image? | 盘子里有什么水果？<br>Which fruit is there in the plate? | 这是什么车？<br>What is the type of the vehicle? | 公交车停在那干吗？<br>Why does the bus park there? |
| GT Answer | 在抓飞盘。<br>Trying to catch the frisbee. | 草原上的马群。<br>Horses on the grassland. | 苹果和橙子。<br>Apples and oranges. | 公交汽车。<br>Bus. | 准备维修。<br>Preparing for repair. |
| mQA Answer | 冲浪。<br>Surfing. | 这是牛。<br>They are buffalos. | 香蕉和橙子。<br>Bananas and oranges. | 火车。<br>Train. | \<OOV\>。<br>\<OOV\> (I do not know.) |

Figure 6: Failure cases of our mQA model on the FM-IQA dataset.

## Footnotes

[1]We are actively developing and expanding the dataset, please find the latest information on the project page : http://idl.baidu.com/FM-IQA.html

[2]The results reported in this paper are obtained from a model trained on the first version of the dataset (a subset of the current version) which contains 120,360 images and 250,569 question-answer pairs.

[3]In practice, we fix the CNN part because the gradient returned from LSTM is very noisy. Finetuning the CNN takes a much longer time than just fixing it, and does not improve the performance significantly.

[4]`http://test.baidu.com`

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
