[Reviews · NeurIPS 2015]

Submitted by Assigned_Reviewer_1

The paper proposes a Turing-test-like evaluation for the visual question answering task. The overall idea has a great potential for the machine learning community and the benchmark dataset, as described, seems to be properly constructed. The proposed baseline for this task is quite convincing and is based on the reuse of a pre-trained model giving latest results from image categorization on Imagenet, as well as the recently proposed neural question answering techniques and from recent natural language generation

techniques based on LSTM.

What is the benefit of non-linearities? The baseline for the visual question answering task involves a lot of non-linear modules that are fully differentiable but maybe simpler baselines (still based on the pre-trained imagenet network), but based on linear layers in the extraction of the question embedding might be interesting to compare with.

Minor comments - references are definitely too small
Summary: The paper is well written, proposes an novel evaluation task for the important problem of visual question answering and develops an end-to-end model combining visual recognition, question understanding and response generation modules that gives a strong baseline for this task.

Submitted by Assigned_Reviewer_2

The paper introduces a novel model of question answering on images using the composition of an LSTM type of model for sentence representation and a CNN type of model for image representation. Both are then combined in a dedicated decision layer for automatic answer prediction. The authors also introduce an interesting crowd generated dataset of visual question answering provided in english and chinese languages. The model contribution is interesting in the sense that it cleverly combines to actual models of the literature for text and image vectorial representation for an up-to-date problem of visual question answering. However, the evaluation is a bit confusing, having the capability to define a proper accuracy of the model would have given to the proposed model and the provided results a bigger impact .
Summary: The paper is clear and well-written, the compositional text+image model is novel and leverage on up-to-date advances in neural machine learning modelization. The dataset creation is also an interesting contribution of the paper. Experimentally speaking, an accuracy measurement would have been preferred to the proposed Turing-test type of evaluation.

Submitted by Assigned_Reviewer_3

The paper describes a new dataset for visual question answering, which is larger that other previously or concurrently developed datasets and contains unconstrained questions about images together with answers. Both Chinese and English versions are planned even though only the Chinese Q-A pairs have been completed by the time of submission.

The dataset promises to be very useful for both the visual QA task and the visual MT task.

A method for visual QA is proposed which is similar to other previously proposed or concurrently developed methods. There are some differences (using separate LSTMs for questions and answers), but their impact is not evaluated.

The quality of the proposed model is high -- the system-generated answers pass a visual Turing test 65% of the time.

P.S. Thanks for the detailed response to my questions.
Summary: The paper describes a new visual QA dataset of unconstrained QA pairs in two languages, which is likely to be widely used. A method very similar to ones used in prior and concurrent work is presented for answering visual questions and the method obtains high accuracy, but there is no analysis / evaluation of the novel aspects of the model. A detailed analysis of the question/answer types occurring in this unconstrained dataset is also lacking.

Submitted by Assigned_Reviewer_4

The paper was fairly clear overall. As stated above, my main concern is that the VTT evaluation might produce very different results from one set of evaluators to another, as is a danger for any such manual evaluation. The scale of resources needed to reproduce it makes it somewhat inaccessible to academic groups. A more detailed experimental protocol, including instructions given to examiners, should be provided.

Aside from that, there are a few few questions/comments for the authors to consider:

* The first part of the title is a little weird. The main/best contribution of the paper is the model. Something more standard might be better.

* Section 3.1 explains the model hyperparameters and architecture, but the selection mechanism was not specified? Why did you use 400 memory cells? Why were the weight matrices of LSTM(Q) and LSTM(A) not shared in the first components?

* 'where "+" denotes element-wise addition" <- as opposed to what??

* Are the top layers of the ConvNet fixed after separate training, or only the convolutional layers?

* Section 3.2 <- in neural language modelling, a common strategy is to (counter-intuitively, perhaps) decouple input and output embeddings, as representations for the purpose of predicting the next symbol may need to encode different information as those modelling the context of the prediction. If tying these matrices (which is also done in many models) provided better validation errors, this should be clearly stated. Otherwise, I'd recommend dropping some of the hand-wavvy fluffiness from this section.

* Section 3.3 <- The initial learning rate is 1 and we decrease it by a factor of 10 for every epoch of the data" <- this sounds massively hacky. How was this procedure selected? Surely not against test results, right?

* "We stop training when the loss does not decrease within three epochs" <- also hacky. How was this procedure selected? Why not use loss on a validation set and some form of patience/early stopping?

* For VTT in section 5.1, what is variance in annotator scoring of models? Is there some form of inter-annotator agreement you could report?

Overall, the idea for the model is not bad, but resembles many LSTM/RNN based QA or sequence to sequence models. My main concern with this paper is that many aspects of the model design and training procedure seem ill-motivated, and make the reported results somewhat hard to judge. The Visual Turing Test is also a problematic means of evaluation, as discussed above. I'd like to see these issues substantially addressed before recommending acceptance.
Summary: This paper provides a multimodal model for combining information from question and image embeddings to answer questions. The architecture is sensible although comparable to similar multimodal systems. The Visual Turing Test evaluation suggested would be very hard to reproduce, especially in academic labs.

Submitted by Assigned_Reviewer_5

1 One issue is there is no baseline model performance from other related work. So it would be good if the paper would include the performance of the current the-state-of-the art model on the introduced data set in order to give us some clarity of the performance of the proposed model.

2 However, the dataset introduced is pretty interesting. The idea of letting user to come up with their own questions based on the image is very innovative and fun, which is much better than the synthesized questions in previous data sets. It would be nice to have a little bit more discussions on how to pre-screen the participants and how to control the quality of the dataset. Providing more examples in the supplementary materials would make the reviewers to have a better understanding of the quality of the dataset.

3 The evaluation process needs some refinements, in Section 5.2, the score 0, 1 and 2 is not well defined in the annotation scheme, which in turn creates confusion. One should not combine two purposes of evaluation in one set of score. In this paper, score 1 is a combination of two things, it is a difficult problem and the answer is partially correct. I would recommend separating the evaluations into two sets of ratings. First, rate the difficulties of the question in the range of 1-3, and then ask whether the answer is ``correct", ``partially" correct or ``incorrect". In addition, it would also be necessary to report the inter-annotator reliabilities of the task.

4 Overall it is a nice work with some flaws. It might be more appropriate to send it to ACL/EMNLP as the novelty of the paper is not the model but the dataset. And the multilingual dataset would attract more attention in NLP field.

5 Typo in Figure3 "on the right hand or left \textbf{hand} side of the gentlemen".
Summary: This is a well-written paper. It introduced a nice multilingual data set for image question answering. The paper used the popular neural work model to combine image and text to perform this task. There are four components in the model, a LSTM for extracting semantic meaning of the question, a deep CNN to extract image concept representations, another LSTM for language sequence generation and finally a component to fuse the previous three models. The approach is not that innovative compared to previous work in image question-answering task. In addition it would be great to put more effort in analysis on why this model captures some of the inferences based on the picture and the question.

Author Feedback
Author rebuttal: We thank the reviewers for their helpful comments. We will incorporate them in the revised manuscript and supp. material. As the reviewers appreciate, we construct an interesting and useful multilingual image Question-Answering (QA) dataset, which will be released. We present a novel and effective model as a strong baseline model of the dataset.

As requested, we provide the comparisons of our model with three additional baselines (see R1, Q3&Q4 of R2). Our model outperforms all of them. We also show the variance and inter-annotator agreement of the proposed Visual Turing Test (VTT) evaluation (see Q2 of R2). We will release the interface of this evaluation on Amazon MTurk.

R1:
Thanks for the suggestion. We implement the baseline which uses linear mapping of bag-of-words of questions as their embeddings. The word error and cost are 0.442 and 2.17 (the lower the better) on val, which are worse than the proposed model (0.393 and 1.91).

R2:
Q1: "The scale of resources needed to reproduce it makes it somewhat inaccessible to academic group"
A: The proposed VTT evaluation is easy to reproduce and costs less than $24 using Amazon MTurk to do one evaluation (2000 QA pairs, 1000 generated ones mixed with 1000 groundtruth ones). We will release the MTurk interface, which includes the detailed instructions to the MTurkers. The cost is affordable to most groups.

Q2: "For VTT in section 5.1, what is variance in annotator scoring of models? Is there some form of inter-annotator agreement...?"
A: As requested, we conduct 3 evaluations with different groups of evaluators. For VTT (Yes/No), the std of the passing rate for Human and mQA model are 0.013 and 0.024 respectively. 3 groups give the same judgement in 95.4% and 92.1% cases for Human and mQA respectively.
For human rated scores of Human and mQA model, the evaluators give the same score in 88.3% and 83.9% cases. The std are 0.020 and 0.041 respectively.

Q3: Using different weight matrices of LSTM(Q)&(A)
A: We implement the baseline where the weights of the two LSTMs are shared. The word error and cost (lower the better) of this baseline are 0.439 and 2.09 on val set, which are worse than the proposed model (0.393 and 1.91).

Q4: Sharing between embedding & softmax
A: The model without this strategy gets a word error and cost of 0.438 and 2.144 on val, which are worse than the proposed model.

Q5: "Why not use loss on a validation set and some form of patience/early stopping?"
A: The "loss" mentioned in line 239 means validation loss as suggested. Sorry for the confusion.

Others: The hyper parameters, including dim of LSTMs and initial learning rate, are selected by cross-validation. The similar learning rate decrease strategy is a common practice for training neural networks [17]. "+" denotes element-wise addition as opposed to concatenation. The top layers of ConvNet are fixed.

R3:
Q1: Comparison with the concurrent model
A: As suggested, we implement [23] on our dataset. Due to the limited time in rebuttal, we only used hyperparameters described in the paper and did not tune them. The loss is 2.86 on val set, which is worse than our model. The reason might be the model structure and hyperparameters that are effective on the DAQUAR dataset used in [23] (which is much smaller than ours) are not suitable on our dataset. We implement three additional baselines and show that our model performs better than all of them (R1, Q3&Q4 of R2).

Q2: Evaluation
A: In the scoring process, we ask the evaluators whether the answer is "correct", "partially correct" or "incorrect". After collecting the results, we find that a few (5.4%) QA pairs with score 1 correspond to questions that is too hard to answer. As suggested, we will rate the difficulties of the questions in both train, val and test sets. The inter-annotator variance and agreement are reported in Q2 of R2.

R4:
We finished both English and Chinese versions of the dataset. There are 9 types of question-answering pairs (e.g. object class, attribute, quantity&number, choice, position, comparison, scene, action and high-level reasoning). We will provide detailed analysis of these types in the main paper and supp. material.

Q1: "There are some differences (using separate LSTMs for questions and answers), but their impact is not evaluated"
A: The model with shared LSTMs weights performs worse than the proposed model on the val set (Q3 of R2). We also provide another two baselines (R1, Q3 of R2).

R6:
Q: "having the capability to define a proper accuracy of the model would have given to the proposed model and the provided results a bigger impact"
A: Thanks for the suggestion. As stated in the first paragraph in Sec. 5, the existing automatic metrics are unsuitable for the image QA task on our dataset. We show that the proposed VTT evaluation is reliable and easy to conduct (Q1&Q2 for R2). We agree that defining a proper accuracy is important and will treat it as future work.